# Coping Strategies and Stress Related Disorders in Patients with COVID-19

**DOI:** 10.3390/brainsci11101287

**Published:** 2021-09-28

**Authors:** Liana Dehelean, Ion Papava, Madalina Iuliana Musat, Mariana Bondrescu, Felix Bratosin, Bianca Oana Bucatos, Ana-Maria Cristina Bortun, Daniela Violeta Mager, Radu Stefan Romosan, Ana-Maria Romosan, Roxana Paczeyka, Talida Georgiana Cut, Silvius Alexandru Pescariu, Ruxandra Laza

**Affiliations:** 1Department of Neurosciences-Psychiatry, “Victor Babes” University of Medicine and Pharmacy, E. Murgu Square, Nr. 2, 300041 Timisoara, Romania; lianadeh@umft.ro (L.D.); mariana.bondescu@umft.ro (M.B.); bianca.bucatos@umft.ro (B.O.B.); ana-maria.bortun@umft.ro (A.-M.C.B.); romosan.radu@gmail.com (R.S.R.); ana.romosan@gmail.com (A.-M.R.); 2Center for Cognitive Research in Neuropsychiatric Pathology, “Victor Babes” University of Medicine and Pharmacy, E. Murgu Square, Nr. 2, 300041 Timisoara, Romania; 3Timis County Emergency Clinical Hospital “Pius Brinzeu”, Liviu Rebreanu, Nr. 156, 300723 Timisoara, Romania; dana.m3@yahoo.com; 4Neuropsychiatry Hospital Craiova—Psychiatry Clinic I, Aleea Potelu 24, 200317 Craiova, Romania; 5Doctoral School, University of Medicine and Pharmacy ‘’Victor Babes’’ Timisoara, E. Murgu Square, Nr. 2, 300041 Timisoara, Romania; talida.cut@umft.ro; 6Department of Infectious Diseases, University of Medicine and Pharmacy ‘’Victor Babes’’ Timisoara, E. Murgu Square, Nr. 2, 300041 Timisoara, Romania; felix.bratosin7@gmail.com (F.B.); ruxi_martincu@yahoo.com (R.L.); 7Clinical Hospital of Infectious Diseases and Pneumophtisiology ‘’Doctor Victor Babes’’ Timisoara, Gheorghe Adam, Nr. 13, 300310 Timisoara, Romania; roxy_klopo@yahoo.com; 8Department VI, Cardiology, University of Medicine and Pharmacy “Victor Babes” Timisoara, E. Murgu Square, Nr. 2, 300041 Timisoara, Romania; pescariu.alexandru@umft.ro

**Keywords:** COVID-19, SARS-CoV-2, coping strategies, acute stress disorder, post-traumatic stress disorder

## Abstract

Patients with severe COVID-19 experience high-stress levels and thus are at risk for developing acute stress disorder (ASD) and/or post-traumatic stress disorder (PTSD). The present study aims to search for correlations between psychiatric response to stress and coping strategies among individuals with acute vs. remitted COVID-19. Ninety subjects with COVID-19 were included in the study, divided into two samples by disease category. Our focus was analysing the perceived stress intensity according to NSESSS and PCL-C-17 scales, and coping strategies with COPE-60. High NSESSS scores were found in 40% of acute patients, and 15.6% of remitted patients had high PCL-C-17 scores fulfilling the criteria for PTSD. We found a negative correlation between stress level and disease category. Acute patients used significantly more engagement and emotion-focused coping methods, but less disengagement types of coping than patients in the remitted phase. Remitted patients under high stress levels are prone to use disengagement and emotion-focused coping strategies. In conclusion, remitted COVID-19 patients experience lower levels of stress and use less emotion-focused strategies, except among those who developed PTSD post-COVID-19 infection, presenting with high-stress levels and using more disengagement and emotion-focused types of coping strategies.

## 1. Introduction

The current pandemic started 12 December 2019, in Wuhan, China, with a group of people suffering from atypical pneumonia. On 11 February 2020, the disease was officially named COVID-19, and the virus SARS-Cov-2 [1]. The new coronavirus is approximately 80% genetically similar to SARS-Cov-1. The similarities between the two viruses provided insight regarding the disease dynamic and pathogenicity, which aided in the adaptation and implementation of disparate prevention measures [2,3,4,5,6,7]. The new coronavirus uses a membrane glycoprotein called spike protein S as a binding receptor to attach to the host cell. Another transmembrane protein-TMPRSS2- triggers protein S to split into two subunits, S1 causing the viral and the host cell membranes to merge. Thus, the virus is endocytosed, and its RNA genome is transcribed to messenger RNA (mRNA). Protein S uses an angiotensin-converting enzyme (ACE2) to bind to the host cell. Consequently, in organs such as lungs, kidneys, heart, and endothelium, where the ACE2 has greater expression, the clinical manifestation of the disease at these sites tends to be more intense [8,9]. Since the actual knowledge of SARS-Cov-2 infection does not fully explain the diversity of its clinical manifestation, further elucidation has been sought in fields such as molecular biology research of the relationship between infection susceptibility and Human-Leukocyte Antigen (HLA), genetic polymorphism of the gene responsible for ACE2 synthesis, Single Nucleotide Polymorphisms (SNPs) implications in proinflammatory and anti-inflammatory cytokines synthesis and the interplay between genetic and -epigenetic mechanisms [5,10,11].

COVID-19 has become a significant burden to healthcare systems, and the economy globally. Between 29 December 2019 and 31 March 2020, 750,890 people were infected with SARS-CoV-2, and 36,405 died of this infection, resulting in a mortality rate of 4.84%. The infection is highly contagious with a large number of people being infected all over the world. Men over 60 years old were the most susceptible to symptomatic infection [12,13]. The coronavirus pandemic is a global health problem with a significant impact on mental health [14]. The social and psychological impact on people suffering from this disease should also be considered for their long-term health effects [15], as some studies demonstrated that good stress response and positive appraisal, specifically of the consequences of the Corona crisis, are the strongest factors in predicting disease trajectory, as compared to other psychological factors [16,17]. At the same time, some individuals seemed to overcome the disease in different manners, some of them developing severe symptoms while others were asymptomatic or mildly ill. Moreover, there were cases of young adults or persons with no clinical history of chronic diseases who developed severe COVID-19 symptoms, whereas others with multiple comorbidities or older individuals overcame COVID-19 more easily. It is well-known from previous outbreaks of infectious diseases that there are consequences not only on the physical body but also on the mind, frequently causing significant psychological distress.

Stress can be explained as a response to a perceived threat, physical and/or psychological. Stress responses are usually accompanied by an emotional response, including anxiety as a central emotion [18]. Researchers have differentiated anxiety into two types regarding duration: state anxiety, which refers to an acute response to possible danger, and trait anxiety, as a chronic state expressed continuously during a person’s life, also being linked to personality [19]. Responses to threats are complex, involving different brain regions such as the hypothalamus, amygdala, prefrontal cortex, and different brainstem nuclei [20]. In regards to these brain regions described to be involved in threat response, some authors have demonstrated that repetitive Transcranial Magnetic Stimulation (rTMS) delivered over the dorsolateral prefrontal cortex (DLPFC) after memory reactivation reduces fear responding to learned fear, bolstering the argument in favour of potential clinical implications for targeting emotional, maladaptive memories [21]. Similarly, the stimulation of rTMS over the DLPFC was illustrated as a better alternative for those patients not responding to psychotherapy and/or drug treatments [22]. Moreover, stimulating the ventromerdial prefrontal cortex (vmPFC) seemed to decrease anxiety and avoidance ratings in patients diagnosed with acrophobia [23].

Therefore, stress and anxiety are linked, which explains why dysregulation in adapting to stressful situations can result in developing certain psychiatric disorders such as acute stress disorder (ASD), post-traumatic stress disorder (PTSD), and adjustment disorder (with anxiety, depression, or both). All of these refer to a specific set of symptoms caused by exposure to a threat, either directly or by learning about a traumatic experienced that a close person was exposed to, according to the Diagnostic and Statistical Manual of Mental Disorders, fifth edition (DSM-5). While acute stress disorder lasts three days to 1 month after exposure to the traumatic event, the post-traumatic stress disorder’s duration has to be longer than one month [24].

When the environment suffers significant changes, people tend to perceive it as a stressful event or danger [25]; thus, they adjust to the new settings using their coping mechanisms. Nevertheless, acute stress seems to interfere differently with the person’s coping strategies compared to chronic stress [26]. COVID-19 generated high levels of stress in the general population due to the short-term and long-term complications of SARS [27]. Parker and collaborators found that 25% of the subjects admitted in nonintensive care wards with COVID-19 had mild-to-moderate acute stress disorder symptoms [28].

In the presence of a stressful event, a person has to deal with the emotions generated by the event. Thus, the coping process has two dimensions, the problem-focused coping strategy and the emotion-focused coping strategy [29,30] or the approach-avoidance coping method (i.e., cognitive efforts to solve the problem of distracting oneself from the problem) and emotional equilibrium-disequilibrium coping strategy (i.e., calming or relaxing oneself versus releasing/suppressing one’s emotions by venting or substance use) [31]. A coping strategy refers to a cognitive, emotional, or behavioural response to stress that is related to a particular function (i.e., problem or emotion-focused coping). A set of coping strategies associated with different functions form a coping mode. Coping strategies may be stable over time (i.e., dimensional/trait-dependent) or not (i.e., dispositional/state-dependent) [32,33]. The coping style refers to stable strategies over time and across different situations [34].

To our knowledge, there is very little data on coping strategies in association with COVID-19. Available studies have evaluated medical staff and patients with chronic medical comorbidities in overcoming the stress related to the pandemic; however, the impact of COVID-19 on stress response among individuals in the general population without chronic medical comorbidities during the acute and remitted phases have yet to be explored [35]. Therefore, this study aimed to describe the most frequent coping strategies or clusters of strategies used by patients with COVID-19 in dealing with the infection and after remission. Firstly, we searched for correlations between the severity of the disease (trauma, stressor), the severity of the psychiatric response (ASD, PTSD), and the clusters of coping strategies. Secondly, we examined factors that may contribute to the overall prognosis in terms of somatic comorbidities, familial status, income, and occupation.

## 2. Materials and Methods

### 2.1. Population

A cross-sectional study was performed on 90 patients admitted to “Victor Babes” Infectious Disease and Pneumophtisiology Hospital of Timisoara from 1 April 2021 to 15 June 2021. Two samples of 45 patients each were established, one with patients diagnosed with COVID-19 infection (acute patient group) and one with COVID-19 remitted patients (post-acute group). The inclusion criteria was comprised of patients diagnosed with SARS-CoV-2 infection by at least one positive Reverse Transcription Polymerase Chain Reaction (RT-PCR) test, patients with evidence of lung damage by thoracic Computed Tomography (CT) evaluation, unimpaired cognitive abilities, consent to participate in the study, and patients presenting clinically significant acute stress disorder symptoms (for the acute patient group). The exclusion criteria were patients under 18 years old, patients in a critical condition unable to understand and answer the questionnaire’s questions, and patients who refused to participate in the study. 

We collected clinical and paraclinical data from both samples, while each sample was divided into three categories by disease severity, as mild, moderate, and severe COVID-19. The mild form is defined by no imagistic signs of pulmonary lesions or mild forms of COVID-19 pneumonia (less than 20% affected pulmonary area), SpO2 above 94% in room air, no oxygen therapy required, and no or mild inflammation syndrome. The moderate severity COVID-19 pneumonia is defined by more than 20% affected pulmonary area but less than 50%, low SpO2 but over 87% in room air, oxygen therapy required for a short period increased inflammation syndrome. Lastly, a severe COVID-19 infection involves more than 50% of the pulmonary area, dyspnea, increased inflammation syndrome, coagulation disorders with platelet counts under 100,000, increased D-dimers, organ lesion signs, and oxygen therapy is required for a long period. All patients underwent treatment according to the Romanian Ministry of Health COVID-19 guidelines at the time of study. Antiviral therapy with Remdesivir, high doses of corticoids, and oxygen therapy were administrated in moderate and severe forms.

To assess the stress level, patients from the acute sample had to fill in The National Stressful Events Survey Acute Stress Disorder Short Scale (NSESSS). Patients from the remitted sample had to answer the PTSD Check List Civilian Version (PCL-C) Questionnaire. Since acute stress disorder develops and resolves within one month from the traumatic event, while PTSD continues beyond the one-month period, it was necessary to use both the NSESSS and PCL-C to evaluate the two different study groups. All subjects also filled up the COPE Questionnaire evaluating the most frequent coping strategies they use.

### 2.2. Ethics

Patients were informed about the study’s aim and implications, each of them having signed a written informed consent after all their questions have been answered. Our study was conducted in agreement with the Helsinki Declaration Guidelines for scientific experiments involving human subjects, and the Scientific Ethics Committee approved it of Infectious Diseases and Pneumophtisiology “Victor Babes” Hospital, Nr.5059 from 28 May 2021.

### 2.3. Scales Interpretation

The severity of the patients’ reaction to the acute stress accompanied by acute COVID-19 infection was assessed using NSESSS. The National Stressful Events Survey Acute Stress Disorder Short Scale (NSESSS) measures seven symptoms/items related to an extremely stressful event. The subjects are asked to rate on a 5-point scale (0 = Not at all; 1 = A little bit; 2 = Moderately; 3 = Quite a bit, and 4 = Extremely) the severity of their symptoms during the past seven days. The total (raw) score ranges from 0 to 28, where a score greater than 14 is attributed to a high-stress level.

To assess the stress level of the remitted patients with COVID-19 infection, we used a 17-item civilian checklist for PTSD (PCL-C-17). The subjects are asked to rate on a 5-point scale (1 = Not at all; 2 = A little bit; 3 = Moderately; 4 = Quite a bit, and 5 = Extreme) the presence and intensity of their symptoms in the past month in response to a stressful event. The score may range from 17 to 85. A score of over 44 suggests the likelihood that the patient meets the diagnostic criteria for PTSD.

COPE Inventory is a 60-item questionnaire assessing 15 coping strategies regarding stress. Every item can be scored from 1 to 4, where one means “I usually don’t do this at all” and four means “I usually do this a lot.” Except for mental disengagement, behavioural disengagement, denial, and substance use, the other coping strategies cannot be labelled as “good” or “bad,” and they are simply different ways to deal with stressful situations. There are several ways of categorizing the coping strategies. One of them refers to the way the person reacts when experiencing stress: engagement versus disengagement. The engagement coping cluster includes those strategies that the individual uses to deal with the stressor or the related emotions such as positive reinterpretation and growth, focus on and venting of emotions, use of instrumental social support, active coping, religious coping, humour, use of emotional/social support, acceptance, suppression of competing activities and planning. By contrast, the coping disengagement cluster refers to those strategies the person uses to escape from stress and its related emotions (mental disengagement, behavioural disengagement, denial, and substance use). When approaching (engaging) or avoiding (disengaging) stress, one may address the stressor itself (problem-focused coping) or its related emotions—the distress (emotion-focused coping). The emotion-focused cluster comprises the following strategies: focus on and venting of emotions, use of instrumental social support, substance use. The problem-focused coping cluster includes strategies such as positive reinterpretation and growth, religious coping, humour, suppression of competing activities, and planning) [36].

Eleven out of fifteen discrete coping strategies are significantly correlated with well-being. In particular, denial and behavioural and mental disengagement have been associated with high levels of stress and low levels of well-being [37]. The questionnaire was validated for the Romanian population, having a good internal consistency expressed by an average Cronbach’s alpha coefficient of 0.74 [38].

### 2.4. Statistics

Descriptive and inferential statistics were performed using the IBM SPSS software version 26.0. Continuous data were represented as mean and standard deviation, while categorical variables were represented as absolute and percentage values. Student’s *t*-test and Mann-Whitney U-test were used for comparing means and median values for continuous and discrete variables, respectively, while the ANOVA and Kruskal-Wallis tests investigated differences in means and median values, respectively, between multiple groups. Spearman’s correlation coefficient was determined for non-parametric variables, while Pearson’s correlation coefficient was used to analyse parametric data. An ordinal regression model was built to observe what factors from disease category, disease severity, and stress level determine the use of a certain coping method. A goodness-of-fit test was performed for statistical analysis of proportions. The significance threshold was set at α = 0.05.

## 3. Results

A total of 90 patients were included in our research, with a 1 to 1 distribution as acute and post-acute patient groups. The majority of patients were men (52.2%), and more than 75% of all individuals involved in the study were parents. The general characteristics of our sample (Table 1) identified an insignificant average age difference between acute and post-acute patients (61 years old vs. 55 years old, respectively). A test of homogeneity for the study populations indicated no statistically significant difference in data distribution for sex, civil status, parental status, and disease severity. 

There is sufficient evidence to conclude that distributions of patients are significantly different by level of income. There were significantly more low-income patients in the acute sample (62.2%) as compared to the post-acute sample (37.8%) (*p*-value = 0.010). Differences regarding age, sex, civil status, or the number of children were not statistically significant between the two samples.

Several comorbidities have been recorded in our samples, such as cardiovascular disease (hypertension, atrial fibrillation, ischemic heart disease), metabolic disease (dyslipidemia, hypercholesterolemia), type II diabetes mellitus, neurologic (Parkinson disease, Alzheimer disease), and neoplasia (bronchopulmonary carcinoma, Hodgkin and non-Hodgkin lymphoma); however only diabetes was highly correlated to COVID-19 infection, being more frequent among the remitted patients (28.9%) (*p*-value < 0.015).

Concerning the stress response, 40% of acute COVID-19 patients had high scores at NSESSS, while 15.6% of patients (*p*-value < 0.009) from the COVID-19 remitted sample fulfilled the criteria for PTSD. 

As for coping strategies, significant statistical differences were found between the samples concerning disengagement, engagement, and emotion-focused coping clusters in those who answered the COPE-60 questionnaire with a score ≥ 3 (Figure 1). Patients used Engagement-focused strategies from the COVID-19 acute sample in 51.1% of the cases compared to 46.6% of COVID-19 remitted patients (*p*-value < 0.044). In what concerns the disengagement coping cluster, fewer patients from the COVID-19 acute sample (40%) used disengagement coping strategies compared to 44.4% from the COVID-19 remitted sample (*p*-value < 0.046). Emotion-focused strategies were used more by acute COVID-19 patients (46.6%) compared to the COVID-19 remitted ones (44.4%) (*p*-value < 0.000), but less than problem-focused strategies 51.1%. On the other hand, remitted patients used more emotion-focused strategies (44.4%) than problem-focused ones (37.8%), but less than those from the acute sample.

As shown in Table 2, a higher level of income was significantly correlated to the remitted patients’ group (r = 0.300, *p*-value = 0.004), while among the studied comorbidities, diabetes was positively correlated to non-acute patients (r = 0.255, *p*-value = 0.015). A negative correlation was significantly found between the level of stress and disease category (r = −0.398, *p*-value < 0.001), thus patients in the acute COVID-19 study group experienced higher stress levels than remitted COVID-19 patients. There were no significant correlation findings regarding coping strategies and the category of the disease. There was a significant association between male gender and disease severity (r = 0.321, *p*-value = 0.002), as well as for old age (r = 0.386, *p*-value <0.001). By civil status, widowed patients were negatively correlated with disease severity (r = −0.226, *p*-value = 0.019). The comorbidities that were tightly associated to severe COVID-19 cases were cardiovascular diseases (r = 0.246, *p*-value = 0.019), and diabetes (r = 0.236, *p*-value = 0.025). Overall, the four clusters of coping strategies that were analyzed here did not have any significant correlation to disease severity.

An ordinal regression model was built to observe what factors out of disease category, disease severity, and stress level determine the use of disengagement, engagement, emotion-focused, and problem focus types of coping (Table 3). The regression model for disengagement was responsible for 9% of the variation (*p*-value = 0.04), although none of the independent variables had a significant independent influence. Though, remitted patients who suffered a severe COVID-19 infection and perceiving high-level stress are prone to use coping disengagement mechanisms. The engagement and the problem focus types of coping were not influenced by the regression model comprising the disease category, disease severity, or stress level (R^2^ = 0.03, *p*-value = 0.44), respectively (R^2^ = 0.02, *p*-value = 0.56). Finally, the regression model for the emotion-focused coping type was statistically significant and accounted for 11% in the variation (R^2^ = 0.11, *p*-value = 0.02), although none of the independent variables had a significant independent influence. However, remitted patients who suffered a severe COVID-19 infection and perceived high-stress levels are prone to use the emotion-focused type of coping.

## 4. Discussion

All four coping clusters (engagement, disengagement, emotion-focused, and problem-focused) were used by both acute and remitted COVID-19 patients. The disengagement and emotion-focused types of coping are more likely to be used by patients who suffered a severe form of infection in the acute state and experienced high levels of stress in their remitted state. Mental or behavioural disengagement, denial, substance use, and venting of emotions are considered maladaptive. During the COVID-19 pandemic, the use of dysfunctional coping strategies was associated with higher levels of anxiety in the general population (self-blame, venting, behavioural disengagement, and self-distraction) [39] and was also described in patients with psychiatric disorders [40,41]. Bailey Holt-Gosselin and collaborators, using the brief COPE, found that women with more severe anxiety symptoms tended to use more maladaptive coping strategies such as self-distraction, denial, venting, substance use, behavioural disengagement, and self-blame during the COVID-19 pandemic [42]. Poor mental health in general during the pandemic was associated with the use of maladaptive coping strategies such as denial, emotional discharge, or substance use [43]. 

In our study, high levels of stress were found in 40% of the acute COVID-19 sample (patients with ASD). At the same time, the prevalence of PTSD in the remitted COVID-19 group was 15.6%. In a meta-analysis, Yuan and collaborators estimated a PTSD prevalence of 23.8% (16.6–31.0%) with COVID-19 patients. They explained its variability by considering factors such as the epidemic particularities and the target population, and the methodology used in the disorder assessment [44]. Later published data show figures in the same range: 22.2%, [45] and 30.2% [46]. 

The type of trauma influences PTSD symptom severity and its onset latency, which is longer in the more severe cases. A study performed on subjects involved in four categories of traumatic events (war veterans, victims of civilian terror, of work-related and of road accidents) showed that the intensity of the PTSD symptoms was higher in war veterans and correlated positively with the time elapsed since the trauma [47]. Another study evaluating the risk of PTSD in patients surviving intensive care showed that 1 in 5 patients developed PTSD symptoms, with an increased risk at 12 months after discharge [48]. In the context of the present pandemic, the incidence of PTSD in SARS-CoV-2 infected patients was higher in the later months of the pandemic, as compared with the first six months (28.8% vs. 18.6%) [44]. We found that the remitted patients who experienced severe forms of COVID-19 infection had higher PCL-C-17 scores (subjects with PTSD) than those with mild or moderate forms of the disease. 

Literature data show that inappropriate coping strategies are listed as risk factors for developing PTSD during the COVID-19 pandemic [43]. In our study, remitted COVID-19 patients diagnosed with PTSD tend to use the disengagement (avoiding stress) cluster strategies. Aldwin and Yancura found that the severity of PTSD was higher in war veterans using avoidant coping strategies and lower when instrumental coping strategies were chosen [49]. There is a relationship between the severity of the trauma, the presence of traumatic dissociation, and the development of PTSD. Several studies [50,51,52] found traumatic dissociation as a predicting factor for developing PTSD. In addition to dissociative symptoms, subjects with PTSD also adopt avoidance as an active defensive strategy to escape from re-experiencing trauma. According to Shalev and collaborators, avoidance appears later in the development of PTSD if trauma extinction does not occur [51]. Following these findings and based on several studies [53,54], a future perspective for studies involving the topic discussed in our research can be an analysis and comparison in biochemical reactions and neurotransmitters implicated in stress adaptation and severity of trauma between patients with acute or remitted COVID-19 infection.

Besides disengagement, our patients with PTSD also used coping strategies from the emotion-focused cluster. This is in accordance with a study performed by Aldwin and Yancura, who found a link between the use of emotion-focused coping and PTSD [49]. Ehring and Quack showed that PTSD symptom severity is significantly associated with certain emotion regulation difficulties such as reduced acceptance of negative emotions, increased experiential avoidance, and suppression of emotions, among others [55].

There is an intricate and biunivocal relationship between neurotransmitters, hormones, and cytokines. While dysregulation at the level of the hypothalamus-pituitary-adrenal axis (HPAA) in PTSD may result in changes in the brain hormones and neuromodulators [56,57], the same neuro-hormonal alterations also result in activation of the peripheral immune response as a part of the fight-or-flight response [58,59]. COVID-19 infection challenges the host immune response, both directly and indirectly. There is literature on the negative impact of the SARS-COV-2 interfering with type 1 interferon signalling. Indirectly, SARS-COV-2 has a negative impact on the immune response through stress response activation of the HPA-axis [60]. High levels of circulating cortisol stimulates humoral immunity and hinders cellular immunity, while norepinephrine through stimulating IL-10, inhibits cellular immunity. Cortisol by inhibiting IL-1, impairs the cellular immunity. Furthermore, altered levels of C-reactive protein, interferon -gamma, interleukin 1-Beta, interleukin-6, and tumour necrosis factor alpha together with variations of the Human Leukocyte Antigen genes (HLA) were found in patients with PTSD [58,61,62,63,64,65]. Significant findings show the importance of inflammation in PTSD [66], with HLA genes playing a major role in neuronal and synaptic plasticity, memory, learning, and behaviour [67,68,69].

Another neurotransmitter/neuromodulator which is affected by the immune response is serotonin. Inflammatory cytokines such as TNF-alpha and INF-gamma stimulate IDO (indoleamine 2,3-oxygenase), an enzyme that converts tryptophane to kynurenine, which may result in a depletion of serotonin and its metabolite, melatonin. Data on animal studies show that selective serotonin reuptake inhibitors, like fluoxetine and paroxetine may increase the brain levels of allopregnanolone [70], a neurosteroid involved in contextual fear extinction [71]. Sustained stress associated with PTSD may predispose to low synaptic levels of serotonin and consequently, depression. Mahar and collaborators found that chronic stress seems to induce a “depressive-like phenotype”, with anhedonia and passive coping mechanisms [72]. Chronic unpredictable mild stress (CUMS) generates behaviour changes lasting more than two weeks after the exposure is ceased [73]. Social isolation may be a consequence of depression or may be an avoidant behaviour in the context COVID-19 PTSD. In this respect, fluoxetine may be a therapeutic strategy to control depression and the response to contextual fear. Lu and collaborators reported that chronic fluoxetine administration in mice prevents depression-like behaviours [74].

Interestingly, patients with PTSD show reduced numbers of naïve CD8+ T-cell and Tregs and increased number of CD3+ and memory T-cells [75], resembling the immune system aging phenotype, thus, making them vulnerable to new viral infections and their somatic and psychological consequences. Considering this, and the association of dysfunctional coping strategies with the vulnerability for PTSD, we advocate for reassurance interventions in acute phase patients, followed by debriefing, cognitive restructuring, and mindfulness techniques as soon as COVID-19 acute patients enter the remitted state to prevent the onset of PTSD.

A limitation of our study arises from the small sample size, limiting the generalizability of results. An important limitation of studies performed on acute patients with COVID-19 experiencing severe forms of infection is that they cannot be properly evaluated with psychiatric questionnaires during the critical state or longitudinally. The lack of control groups also limits our results, since the general population unaffected by SARS-CoV-2 infection would also be psychologically affected by the pandemic in general, despite not being infected.

## 5. Conclusions

Patients in the acute COVID-19 group were exposed to higher stress levels than remitted ones. The disengagement type of coping was used more frequently by remitted COVID-19 patients, while those with acute COVID-19 infection tended to use mostly engagement and emotion-focused strategies. However, we found no significant correlations between the disease category group and the four clusters of coping. Subjects with severe forms of COVID-19 that in the remitted state experienced high levels of stress are prone to use the disengagement and emotion-focused types of coping more often. The comorbidities that were tightly associated with severe COVID-19 cases were cardiovascular diseases and diabetes.

Due to the intricate relationships between central nervous, endocrine, and immune systems, when treating COVID-19, one must actively target the psychological response of the individual, which may be either functional (positive appraisal) or dysfunctional (disengagement). Screening for dysfunctional coping strategies in patients with COVID-19 infection, monitoring, providing reassurance and specific psychological interventions as soon as it is possible, may reduce patients’ vulnerability for developing PTSD. In the long term, sustained high cortisol levels may result in chronic depression and possible cognitive decline.

Lastly, we believe the current research supports the direction for future studies to follow-up patients with acute COVID-19 infection, especially those with high levels of acute stress who may be at risk to develop PTSD and adjustment disorder, while also reassessing the patients to identify which of their coping strategies remain stable over time.

## Figures and Tables

**Figure 1 brainsci-11-01287-f001:**
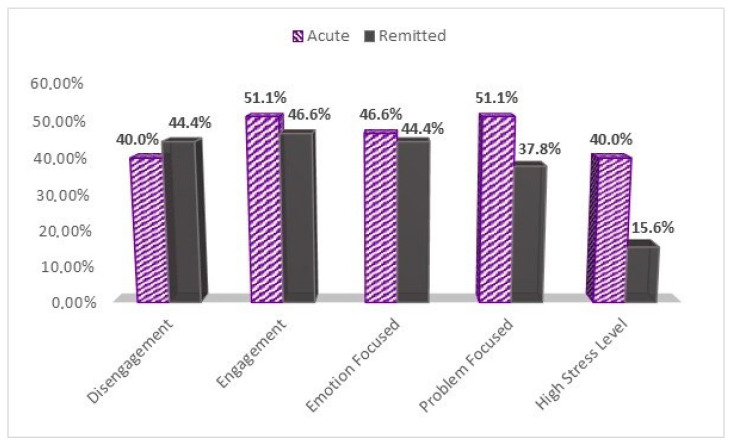
High level of stress and coping strategies between the study groups. Graphical description of patients’ proportions scoring ≥ 3 on the COPE-60 questionnaire, and higher than the median score of 14 on the NSESSS, or higher than 44 on the PCL-C to assess high stress levels.

**Table 1 brainsci-11-01287-t001:** General characteristics of the study groups. Data in this table was divided by disease category to compare the variables describing general characteristics and survey results.

Sample Characteristics		Disease Category		*p*-Value
		Acute (*n* = 45)	Remitted (*n* = 45)	
Age (mean)		61 ± 14	55 ± 15	0.073
Sex				0.205
Male		20(42.5%)	27(57.5%)	
Female		25(58.1%)	18(41.9%)	
Civil Status				0.398
Married		34(75.6%)	35(77.8%)	
Single		4(8.9%)	3(6.7%)	
Divorced		1(2.2%)	4(8.9%)	
Widowed		6(13.3%)	3(6.7%)	
Children				0.334
Yes		38(84.4%)	41(91.1%)	
No		7(15.6%)	4(8.9%)	
Income				0.010
Low		28(62.2%)	17(37.8%)	
Medium		12(26.7%)	11(24.4%)	
High		5(11.1%)	17(37.8%)	
Disease Severity				0.837
Mild		9(20.0%)	7(15.6%)	
Moderate		18(40.0%)	20(44.4%)	
Severe		18(40.0%)	18(40.0%)	
Comorbidities				
Cardiovascular		25(55.6%)	18(40.0%)	0.103
Diabetes		4(8.9%)	13(28.9%)	0.015
Metabolic		7(15.6%)	8(17.8%)	0.500
Neurologic		3(6.7%)	1(2.2%)	0.308
Neoplasia		4(8.9%)	1(2.2%)	0.180
**Survey/Scale**		NSESSS-Total (>14)	PCL-Total (>44)	<0.009
Low-stress		27(60.0%)	38(84.4%)	
High-Stress		18(40.0%)	7(15.6%)	
COPE-60				
Disengagement (2)	>Median	18(40%)	20(44.4%)	0.046
	≤Median	27(60%)	25(55.6%)	
Engagement (3)	>Median	23(51.1%)	21(46.6%)	0.044
	≤Median	22(48.8%)	24(53.3%)	
Emotion Focused (2)	>Median	21(46.6%)	20(44.4%)	<0.000
	≤Median	24(53.3%)	25(55.6%)	
Problem Focused (3)	>Median	23(51.1%)	17(37.8%)	0.125
	≤Median	22(48.8%)	28(62.2%)	

**Table 2 brainsci-11-01287-t002:** Correlation analysis between disease category or disease severity and general characteristics, comorbidities, and coping strategies.

	Disease Category (Acute/Remitted)		Disease Severity	
	Correlation	*p*-Value	Correlation	*p*-Value
Sex(male)	0.156	0.143	0.321	0.002
Age	−0.186	0.080	0.386	<0.001
Civil Status	−0.034	0.750	−0.226	0.032
Children	0.102	0.340	0.047	0.663
Income	0.300	0.004	−0.117	0.270
Cardiovascular	−0.156	0.143	0.246	0.019
Diabetes	0.255	0.015	0.236	0.025
Metabolic Disease	0.030	0.780	0.043	0.684
Neurologic	−0.108	0.312	0.076	0.475
Neoplasia	−0.146	0.171	0.123	0.247
Stress	−0.398	<0.001	0.035	0.743
Disengagement	−0.078	0.468	0.047	0.661
Engagement	−0.057	0.591	−0.152	0.154
Emotion-Focused	−0.111	0.296	−0.173	0.103
Problem Focus	−0.112	0.293	−0.063	0.558

**Table 3 brainsci-11-01287-t003:** Regression analysis by coping strategy Ordinal regression model to observe if disease category, disease severity, and stress level determine the use of disengagement, engagement, emotion-focused, and problem focus types of coping.

	Disengagement				Engagement				Emotion-Focused				Problem Focus			
	R^2^	*p*	CI		R^2^	*p*	CI		R^2^	*p*	CI		R^2^	*p*	CI	
Disease Category		0.69	−0.62	0.94		0.58	−1.01	0.56		0.83	−0.86	0.69		0.21	−1.28	0.28
Disease Severity	0.09	0.69	−0.39	0.59	0.03	0.11	−0.90	0.09	0.11	0.04	−1.01	−0.09	0.02	0.48	−0.67	0.31
Stress Level	(0.04) *	0.01	0.32	2.02	(0.44) *	0.85	−0.74	0.90	(0.02) *	0.02	0.17	1.85	(0.56) *	0.53	−1.08	0.56

* Regression model *p*-value.

## Data Availability

The data presented in this study are available on request from the corresponding author.

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
