# Peer review of "Coping Strategies and Stress Related Disorders in Patients with COVID-19"

_brainsci, 2021, doi:10.3390/brainsci11101287_

Round 1
Reviewer 1 Report
In the present study entitled ‘Coping Strategies and Stress Related Disorders in Patients with COVID-19', by Dehelean and colleagues, authors aimed to examine coping strategies that patients with COVID-19 used to deal during the infection and after remission. For this purpose, the study was performed on 90 patients, divided in ‘acute patient group’ and ‘post-acute group’; to assess the stress level, patients from the acute group completed the National Stressful Events Survey Acute Stress Disorder Short Scale (NSESSS), patients from the remitted group answered the PTSD Check List Civilian Version (PCL-C) Questionnaire. All patients also filled up the COPE Questionnaire, that evaluates the most frequent coping strategies they use. Results showed high NSESS scores in 40% of acute patients, while 15.6% of remitted patients had high PCL-C-17 scores, fulfilling the criteria for PTSD. Acute patients used more emotion-focused coping methods than patients in the remitted group, while remitted patients are more likely to use disengagement types of coping.
In general, I think the idea of the study is interesting and the fascinating results that the authors have found might be of interest to the readers of Brain Sciences. However, some minor comments, as well as some crucial citations that should be included to support the authors’ claim, need to be addressed to improve the research article and its readability prior the publication in the present form.
Minor comments:
- Page 2, lines 76-78: Authors stated that ‘The social and psychological impact on people suffering from this disease should also be considered for their long-term health effects’. In this regard, I suggest adding some key references that would be essential in this section, such as Kunzler and colleagues’ review (2021, Globalization and health), in which authors examine mental health impact of the COVID-19 pandemic in the general population, healthcare workers and patients. Also, in a key study, Veer and colleagues (2021, Translational Psychiatry), tested nearly 16 thousand European adults during the most intense phase of lockdown in Europe and showed that participant’s mental resilience, as an outcome of the perceived experiences and conceptualized as good mental health despite CODIV-19 stressor, was mediated by the ability to easily recover from stress. Thereby, it is demonstrated that good stress response and positive appraisal, specifically of the consequences of the Corona crisis, are the strongest factors as compared to other psychological factors.
- Page 2-3, Introduction: Authors described in detail the etiology of stress disorders and explain how individuals deal with stressful events using coping strategies. I think that this section would benefit from some evidence about new treatments for such debilitating disorders, such as the implementation of non-invasive brain stimulation (NIBS). Thus, I would suggest to add some key references that highlight how NIBS are broadly involved in the study and in the treatment of different psychiatric disorders, like post-traumatic stress disorder (PTSD), anxiety and phobia. In this regard, a recent study by Borgomaneri and colleagues (2020, Current Biology) might be of interest. In this study, authors showed how repetitive Transcranial Magnetic Stimulation (rTMS) delivered over the dorsolateral prefrontal cortex (DLPFC) after memory reactivation reduces fear responding to learned fear, arguing in favor of potential clinical implications for targeting emotional, maladaptive memories. Hermann and colleagues (2017, Brain Stimulation) used rTMS over the ventromedial prefrontal cortex (vmPFC) before a virtual reality exposure to heights participants diagnosed with acrophobia, and found decreased anxiety and avoidance ratings. Furthermore, Borgomaneri and colleagues (2021, Journal of Affective Disorders) illustrated the therapeutic potential of rTMS over DLPFC as a valid alternative for those patients not responding to psychotherapy and/or drug treatments.
- Page 8, lines 308-312: Authors stated that ‘the use of dysfunctional coping strategies...was also described in patients with psychiatric disorders’. In this regard, I would suggest adding two findings that provide additional information on how inappropriate and pathological behaviors could be influenced by maladaptive cue-guided choices, and on how individual differences in learning style or cognitive abilities play a role in the predisposition to such maladaptive implications: decisive evidence from Garofalo and colleagues’ study (2019, Scientific Reports), in which 100 healthy participants performed a decision-making and learning task, suggest that individual differences in working memory capacity are correlated to the specific individuals’ behavior requested by the task and guide behavior toward the most convenient choice. Also, results from a recent study by the same research group (Garofalo et al., 2021, Cortex) show causal evidence for the involvement of the dorsolateral prefrontal cortex (DLPFC) in two forms of human cue-guided choice, namely outcome-specific and general, having crucial implication in addiction and dependence.
- In my opinion, I think the “Conclusion” paragraph would benefit from some thoughtful as well as in-depth considerations by the authors, because as it stands, it is very descriptive but not enough theoretical as a discussion should be. Authors should make an effort, trying to explain the theoretical implication as well as the translational application of their research.
- Regarding the abstract: according to the Journal’s guidelines, authors should have provided an abstract of about 200 words maximum. Indeed, the current one includes 238 words.
- Page 5, lines 232-233: I suggest rewriting this sentence for clarity. Regarding the figure and the tables: I ask authors to check again the Journal’s guidelines and add an explanatory caption for each one. Also, I suggest to modify the graph type showed in Figure 1 to a clustered column chart, in order to improve the clarity of the results.
- According to the Journal’s guidelines, the authors should have provided a link to the data set, where the experimental data are deposited, in order to ensure the replicability of the study.
- Page 10, lines 423-424: provide the abbreviated journal name in italics, the year of publication in bold and the volume number in italics to the following citation ‘Marincu I, Bratosin F, Vidican I, Bostanaru AC, Frent S, Cerbu B, Turaiche M, Tirnea L, Timircan M. Predictive Value of Comorbid Conditions for COVID-19 Mortality. Journal of Clinical Medicine. 2021 Jan;10(12):2652’.
- Page 11, lines 437-438: provide the publisher’s name and the publisher location to the following citation ‘Association, A. P. Trauma- and Stressor-Related Disorders. In Diagnostic and Statistical Manual of Mental Disorders fifth edition; 2013; pp 271–286’.
- Page 11, line 449: provide the publisher’s name and the publisher location to the following citation ‘Lazarus, R. S. F. S. Cognitive Appraisal Processes. In Stress, Appraisal, and Coping; 1984; pp 22–52’.
- Page 12, lines 500-501: provide the abbreviated journal name in italics, the year of publication in bold and the volume number in italics to the following citation ‘M Sherin JE, Nemeroff CB. Post-traumatic stress disorder: the neurobiological impact of psychological trauma. Dialogues Clin 2011;13(3):263-278. doi:10.31887/DCNS.2011.13.2/jsherin’.
- Page 12, lines 485-486: correct the year of publication to the following citation ‘Amir, M.; Kaplan, Z.; Kotler, M. Type of Trauma, Severity of Posttraumatic Stress Disorder Core Symptoms, and Associated Features. J. Gen. Psychol. 2010, 123 (4), 341–351. https://doi.org/10.1080/00221309.1996.9921286’.
- Page 12, lines 490-491: provide the publisher’s name and the publisher location to the following citation ‘Aldwin, C. M.; Yancura, L. A. Coping and Health: A Comparison of Hte Stress and Trauma Literatures. In Physical Health Consequences of Exposure to Extreme Stress; 2004; pp 2–56’.
- Page 12, lines 496-497: correct the authors name and provide the last name followed by the initials of the first name to the following citation ‘A.Y. Shalev, T. Peri, L. C. S. S. Predictors of PTSD in Injured Trauma Survivors: A Prospective Study. Am J Psychiatry 1996, 153 496, 219–225’.
- Page 12, lines 498-499: provide the book title in italics, the publisher’s name and the publisher location to the following citation ‘Nussbaum, L., Hogea, L., Folescu, R., GrigoraÈ™, M., Zamfir, C., Boancă, M., Erdelean, D., et. al. Biochemical Modifications Study of Cerebral Metabolites by Spectroscopy in Epilepsy Treatment. Revista De Chimie, 2018; Volume 69, pp. 965-970'.
Author Response
please find in the attachment.

Reviewer 2 Report
The authors studied coping strategies and stress-related disorders in patients with COVID-19. The findings are interesting and the manuscript is well-organized. I only have a few minor comments.
1. The central hypothesis of this manuscript is unclear. Are the authors making the comparison between COVID-19 vs. other diseases or between COVID-19 (as one disease type) vs. health control? In either way, control groups are missng in order to make any comparison.
2. Please discuss the potential influence of immune activation induced by COVID-19 on stress copying mechanisms.
3. It would be interesting if a general discussion on how these can relate to the changes of neuromodulatory signals due to the COVID-19(social isolation). Such as serotonergic signaling during stress (https://doi.org/10.1016/j.neulet.2019.02.022).
4. Small grammatical errors are seen. Please double-checkAuthor Response
please find in the attachment
